# Disrupted rhythms of life, work and entertainment and their associations with psychological impacts under the stress of the COVID-19 pandemic: A survey in 5854 Chinese people with different sociodemographic backgrounds

**Min Yang[1][ʘ], Ping He[2][ʘ], Xiaoming Xu[3][ʘ], Dan Li[4][ʘ], Jing Wang[1], Yanjun Wang[1], Bin Wang[1], Wo Wang[5], Mei Zhao[4], Hui Lin[6], Mingming Deng[7][‡]\*, Tianwei Deng[8][‡]\*, Li Kuang[3][‡]\*, Dongfeng Chen[1][‡]\***

**1** Army Medical Center of PLA, Daping Hospital, Army Medical University, Chongqing, P.R. China, **2** Department of Gastroenterology, Yongchuan Hospital of Chongqing Medical University, Chongqing, P.R. China, **3** Department of Psychiatry, the First Affiliated Hospital of Chongqing Medical University, Chongqing, P.R. China, **4** CAS Key Laboratory of Mental Health, Institute of Psychology, Beijing, P.R. China, **5** Mental Health Center, University-Town Hospital of Chongqing Medical University, Chongqing, P.R. China, **6** Department of Statistics, Army Medical University, Chongqing, P.R. China, **7** Department of Gastroenterology, The Affiliated Hospital of Southwest Medical University, Luzhou, P.R. China, **8** Department of Gastroenterology, Three Gorges Hospital of Chongqing University, Chongqing, P.R. China

ʘ These authors contributed equally to this work.
‡ These authors also contributed equally to this work.
\* chendf1981@126.com (DC); kuangli0308@163.com (LK); dengtw1986@163.com (TD); 793070544@qq.com (MD)

## Abstract

### Background & aim

The coronavirus disease 2019 (COVID-19) pandemic has affected the life and work of people worldwide. The present study aimed to evaluate the rhythm disruptions of life, work, and entertainment, and their associations with the psychological impacts during the initial phase of the COVID-19 pandemic.

### Method

A cross-sectional study was conducted from the 10th to 17th March 2020 in China. A structured e-questionnaire containing general information, the Chinese version of Brief Social Rhythm Scale, and Zung's self-rating scales of depression and anxiety (SDS and SAS) was posted and collected online through a public media (*i.e.* EQxiu online questionnaire platform). Scores in sleeping, getting up, and socializing (SGS) rhythm and eating, physical practice, and entertainment (EPE) rhythm were compared among and between participants with different sociodemographic backgrounds including gender, age, education, current

**Data Availability Statement:** All relevant data are within the manuscript and its Supporting Information files.

**Funding:** This study was supported by the grant from Chongqing basic research and foreword exploration project, China (Grant No. CSTC2018jcyjAX0620); and Chongqing Medical University, a special project of the emergency clinical research on the new coronavirus disease (general project): (1)Psychological intervention for first-line medical personnel in the new coronavirus pneumonia pandemic, and (2) psychological intervention model of negative emotion and behavior among hospital workers during the COVID-19 pandemic.

**Competing interests:** The authors have declared that no competing interests exist.

occupation, annual income, health status, and chronic disease status. Correlations of SDS and SAS with SGS-scale and EPE-scale were also analyzed.

## Results

Overall, 5854 participants were included. There were significant differences in the scores of SGS-scale and EPE-scale among people with different sociodemographic backgrounds. The scores were significantly higher in the groups with female gender, low education level, lower or higher than average income, poor health status, ages of 26–30 years or older than 61 years, nurses and subjects with divorce or widow status. There were also significant differences in SAS and SDS scores among people with different sociodemographic backgrounds (all $P$< 0.05). The overall prevalence of depression and anxiety was 24.3% and 12.6%, respectively, with nurses having the highest rates of depression (32.94%) and anxiety (18.98%) among the different occupational groups. SGS-scale was moderately correlated with SDS and SAS, and disruption of SGS rhythm was an independent risk factor for depression and anxiety.

## Conclusion

Social rhythm disruption was independently associated with depression and anxiety. Interventions should be applied to people vulnerable to the rhythm disruption during the COVID-19 pandemic.

## Introduction

Coronavirus disease 2019 (COVID-19) is an acute respiratory infectious disease caused by a novel coronavirus, called severe acute respiratory syndrome coronavirus 2 (SARS-CoV-2) [1–3]. Due to the strong infectivity of the coronavirus and the lack of effective treatments and vaccines, COVID-19 has spread rapidly and become a pandemic worldwide with serious consequences. The social restriction policies implemented to control the infection in almost all countries have resulted in unemployment, financial hardship, and a reduction in quality of life in many countries [4], which have seriously affected the life and work of people. Simultaneously, the COVID-19 pandemic has significant implications for a global increase in mental health problems, and the psychological burden during the COVID-19 pandemic has been reported [5, 6].

Social rhythm is defined as the regularity of timing of the behaviors in social activities and also described clinically as "daily routines" [7]. COVID-19-related social changes, particularly lockdown protocols, have indeed impacted the timing of daily behaviors in stabilizing circadian function, and social rhythm disruption may be correlated with negative psychological impacts [7]. Previous studies have reported that social rhythm disruption is connected with mental diseases, especially depression and bipolar disorder in the non-pandemic days, which has been explained by many theories [8–11]. Recently, several studies have reported social and circadian rhythm disruptions in students, office workers, the elderly, and patients with narcolepsy or autism [12–14]. However, so far it is not clear the characteristics of the disrupted rhythms of life, work and entertainment behaviors with different sociodemographic backgrounds and their associations with psychological impacts under the stress of the COVID-19 pandemic. Therefore, the aim of the present study was to evaluate the rhythm disruptions in

life, work, and entertainment and their associations with the psychological impacts in the general Chinese population with different sociodemographic backgrounds during the initial phase of the COVID-19 pandemic.

## Materials and methods

### Participants and procedure

Chinese people with different sociodemographic backgrounds voluntarily participated in this online survey. An e-questionnaire was designed to investigate the rhythm activities and psychological impacts in Chinese people with different backgrounds including gender, age, educational background, annual income, health status, current occupation, and chronic disease status during the COVID-19 pandemic through EQxiu online questionnaire platform. The survey was conducted from 10–17 March 2020 in 22 provinces and five autonomous regions, four municipalities, and two special administrative regions (*i.e.*, Hong Kong and Macao) of China. The study protocol was carried out following relevant guidelines and regulations and approved by the Ethics Committee of the Army Medical Center (Approval No. 2019–144). During the survey, the participants were asked to read and check the informed consent and then complete the questionnaire; those under 18 years old were asked to consent by their guardians before completing the questionnaire. All completed questionnaires were checked for validity and completeness.

### Measures

**Social rhythm.** The Brief Social Rhythm Scale (BSRS) was developed to quickly assess social rhythmicity in large-scale multi-national samples and multiple languages [15]. The Chinese version of BSRS also included entertainment and physical practice, and was shown to have good reliability and validity in China [16]. The scale consisting of 17 items was used to investigate an individual's rhythm in six aspects, i.*e*., sleeping (going to bed), getting up, socializing (SGS), eating, physical practice, and entertainment (EPE) on weekdays and weekends. The first eight items addressing SGS rhythm constituted SGS-scale, and the last nine items addressing EPE rhythm constituted EPE-scale. Each item was assigned with a score of 1, 2, 3, 4, 5, or 6, corresponding to categories of very regularly, quite regularly, somewhat regularly, somewhat irregularly, quite irregularly, or very irregularly, respectively. The lower the score, the more regular the rhythm.

**Zung's depression and anxiety scale.** Zung's self-rating depression scale (SDS) and Zung's self-rating anxiety scale (SAS) were used to evaluate the psychological characteristics of each participant [17, 18]. There were 20 items in SAS and SDS, and each item was rated on a scale of 1–4 reflecting never, often, sometimes, and always. The raw score was the sum of 20 questions multiplied by 1.25 and rounded to get the standard score as described in previous studies [19]. The cut-off value was 50 for both SAS and SDS. The higher the score, the more obvious the anxiety or depression was, with the scores of 50–59, 60–69, and 70 points or more indicating mild, moderate, and severe anxiety or depressive symptom, respectively. The Chinese version of SDS and SAS, which was used in the present study, was reported to present good reliability, validity and excellent internal consistency, with the Cronbach coefficient of 0.784 for SDS in Chinese people [20], and of 0.931 for SAS in the general population [21].

### Statistical analysis

Data were analyzed using SPSS version 23.0 (SPSS, Chicago, IL, USA). The data of normal distribution was expressed by mean±standard deviation. One-way analysis of variance with a

post-hoc analysis, was used to make comparisons among groups. T-test or a nonparametric test (*i.e.*, Mann-Whitney U test or Kruskal-Wallis test), where appropriate, was used to make comparisons between the group with the highest score and each of the other groups for each item. Pearson correlation or Spearman correlation, where appropriate, was used to determine the correlation between the BSRS scales and the Zung's scales. In addition, a multivariate linear regression analysis was used to determine the independent factors for the correlation of anxiety or depression with social rhythm. Cohen's d, an indication of effect size, was generated and reported to further indicate the strength of the difference. A *P* value of < 0.05 was considered statistically significant (corrected for multiple comparisons where appropriate).

## Results

### Sociodemographic characteristics of the participants

A total of 5872 people checked the e-questionnaire. Eighteen questionnaires with invalid data were excluded due to frivolous or incomplete answers, or the same option for different questions or orderly answers. The data of 5854 participants with valid questionnaires were further analyzed. The general information on the sociodemographic characteristics of the participants is listed in Table 1.

### Disruptions of social rhythm in the participants

Generally, there were significant differences in SGS-scale and EPE-scale among participants with different sociodemographic backgrounds including gender, age, educational background, annual income, health status, current occupation, and chronic disease status (*P*<0.05, Table 2). Moreover, the SGS-scale and EPE-scale scores were compared between the group

**Table 1. General sociodemographic characteristics of 5854 analyzed participants.**

| Feature | | N (%) | Feature | | N (%) |
|---|---|---|---|---|---|
| Gender | Female | 3439(58.75%) | Health status | Poor | 53(0.91%) |
| | Male | 2415(41.25%) | | Normal | 787(13.44%) |
| Age | -18 | 68(1.16%) | | Good | 3686(62.97%) |
| | 18–25 | 1960(33.48%) | | Very good | 1328(22.69%) |
| | 26–30 | 862(14.72%) | Current occupation | Businessman | 481(8.22%) |
| | 31–40 | 1339(22.87%) | | Officer | 245(4.19%) |
| | 41–50 | 852(14.55%) | | Teacher | 231(3.95%) |
| | 51–60 | 682(11.65%) | | Police | 277(4.73%) |
| | 61- | 91(1.55%) | | Farmer | 217(3.71%) |
| Marital status | Unmarried | 2435(41.60%) | | Employee | 270(4.61%) |
| | Married | 3232(55.21%) | | Doctor | 1171(20.00%) |
| | Divorced /widowed | 187(3.19%) | | Nurse | 1017(17.37%) |
| Chronic disease | No chronic disease | 4373(74.70%) | | Medical technician | 139(2.37%) |
| | CDCPD | 671(11.46%) | | Retired re-employee | 229(3.91%) |
| | Chronic diseases only | 810(13.84%) | | Non-medical student | 1172(20.02%) |
| Annual income | Less than 50000 CY | 2482(42.40%) | | Medical student | 405(6.92%) |
| | 50000–100000 CY | 1849(31.59%) | Education | Junior high school or lower | 97(1.66%) |
| | 100000–200000 CY | 1188(20.29%) | | Senior high school | 535(9.14%) |
| | 200000–300000 CY | 225(3.84%) | | Junior college or undergraduate | 4465(76.27%) |
| | More than 300000 CY | 110(1.88%) | | Master or above | 757(12.93%) |

CY, Chinese yuan; CDCPD, chronic diseases comorbid with psychosomatic diseases.

**Table 2. Scores of social rhythms in participants with different sociodemographic backgrounds during the COVID-19 pandemic (N = 5854).**

| Feature | | SGS-scale | | | | EPE-scale | | | |
|---|---|---|---|---|---|---|---|---|---|
| | | Mean±SD | Z/X2 | P | Cohen's d | Mean±SD | Z/X2 | P value | Cohen's d |
| Gender | Female | 21.64±9.03 | -10.183 | <0.001 | 0.269 | 7.71±6.02 | -7.627 | <0.001 | - |
| | Male | 19.24±8.80 | | | - | 8.71±5.93 | | | 0.167 |
| Age | -18 | 17.63±8.69 | 146.237 | <0.001 | - | 7.82±5.58 | 112.103 | <0.001 | 0.102 |
| | 18–25 | 18.98±8.95 | | | 0.153 | 7.75±5.35 | | | 0.091 |
| | 26–30 | 23.04±9.77 | | | 0.585 | 7.22±6.22 | | | - |
| | 31–40 | 21.54±8.95 | | | 0.443 | 7.76±6.18 | | | 0.087 |
| | 41–50 | 20.79±8.56 | | | 0.366 | 9.10±6.68 | | | 0.291 |
| | 51–60 | 20.74±7.91 | | | 0.374 | 9.64±5.82 | | | 0.402 |
| | 61- | 21.27±8.78 | | | 0.417 | 9.87±6.14 | | | 0.429 |
| Marital status | Unmarried | 19.53±9.19 | 81.065 | <0.001 | - | 7.81±5.52 | 2.499 | 0.294 | - |
| | Married | 21.42±8.77 | | | 0.210 | 8.32±6.26 | | | 0.086 |
| | Divorced /widowed | 22.04±9.21 | | | 0.273 | 8.76±7.25 | | | 0.147 |
| Education | Junior high school or lower | 18.92±10.07 | 8.880 | 0.031 | - | 7.81±7.18 | 11.172 | 0.011 | 0.009 |
| | Senior high school | 21.18±8.97 | | | 0.237 | 7.75±5.97 | | | - |
| | Junior college or undergraduate | 20.64±9.06 | | | 0.180 | 8.05±5.89 | | | 0.051 |
| | Master or above | 20.59±8.61 | | | 0.178 | 8.85±6.46 | | | 0.177 |
| Annual income | Less than 50000 CY | 19.21±8.90 | 126.387 | <0.001 | - | 7.88±5.64 | 48.455 | <0.001 | 0.034 |
| | 50000–100000 CY | 22.18±9.26 | | | 0.327 | 7.68±6.06 | | | - |
| | 100000–200000 CY | 21.28±8.49 | | | 0.238 | 8.93±6.28 | | | 0.203 |
| | 200000–300000 CY | 20.66±8.41 | | | 0.167 | 9.31±6.03 | | | 0.270 |
| | More than 300000 CY | 20.78±9.03 | | | 0.175 | 9.86±8.12 | | | 0.304 |
| Health status | Poor | 28.02±10.00 | 500.564 | <0.001 | 1.178 | 9.53±8.85 | 26.969 | <0.001 | 0.274 |
| | Normal | 25.21±9.48 | | | 0.906 | 7.94±7.19 | | | 0.065 |
| | Good | 20.93±8.36 | | | 0.468 | 8.35±5.90 | | | 0.146 |
| | Very good | 16.90±8.85 | | | - | 7.53±5.30 | | | - |
| Current occupation | Businessman | 20.21±8.50 | 322.663 | <0.001 | 0.302 | 8.39±5.92 | 177.273 | <0.001 | 0.322 |
| | Officer | 19.89±8.18 | | | 0.269 | 9.32±5.94 | | | 0.476 |
| | Teacher | 21.06±8.41 | | | 0.404 | 9.74±6.77 | | | 0.510 |
| | Police | 18.69±8.79 | | | 0.121 | 9.05±5.46 | | | 0.448 |
| | Farmer | 19.81±9.55 | | | 0.239 | 7.47±6.17 | | | 0.165 |
| | Employee | 20.49±7.99 | | | 0.345 | 8.44±5.81 | | | 0.333 |
| | Doctor | 20.99±8.27 | | | 0.399 | 8.52±6.39 | | | 0.330 |
| | Nurse | 24.47±9.91 | | | 0.740 | 6.46±6.08 | | | - |
| | Medical technician | 21.31±8.65 | | | 0.428 | 7.63±6.56 | | | 0.185 |
| | Retired re-employee | 21.63±8.78 | | | 0.462 | 9.62±5.28 | | | 0.555 |
| | Non-medical student | 17.65±8.46 | | | - | 7.90±5.20 | | | 0.255 |
| | Medical student | 20.64±8.80 | | | 0.346 | 8.66±6.08 | | | 0.362 |
| Chronic disease | No chronic disease | 19.71±8.82 | 202.421 | <0.001 | - | 7.94±5.67 | 6.269 | 0.044 | - |
| | CDCPD | 24.01±9.18 | | | 0.478 | 8.67±7.40 | | | 0.111 |
| | Chronic diseases only | 22.98±8.82 | | | 0.371 | 8.68±6.37 | | | 0.123 |

SD, standard deviation; CY, Chinese yuan; SGS-scale: the rhythm of sleep, getting-up, and socializing; EPE-scale: the rhythm of eating, physical practice, and entertainment activities; CDCPD, Chronic diseases comorbid with psychosomatic diseases.

with the highest score and each of other groups (Table 3). Specifically, the female gender and poor health status were most closely associated with disrupted rhythms of life, work, and entertainment. The age group of 26–30 years, nurses and subjects with divorce or widow status, the

**Table 3. Comparison of SGS-scale and EPE-scale scores between the group with the highest score and each of the other groups in terms of sociodemographic backgrounds.**

| Features | SGS-scale | | | | EPE-scale | | | |
|---|---|---|---|---|---|---|---|---|
| | Highest-score group | Other groups | P-value | Cohen's d | Highest-score group | Other groups | P-value | Cohen's d |
| Marital status | Divorced /widowed | Unmarried | 0.001 | 0.273 | Divorced /widowed | Unmarried | 0.227 | 0.147 |
| | | Married | 0.748 | 0.069 | | Married | 0.802 | 0.065 |
| Chronic disease | CDCPD | No chronic disease | <0.001 | 0.478 | Chronic diseases only | No chronic disease | 0.006 | 0.123 |
| | | Chronic diseases only | 0.086 | 0.114 | | CDCPD | 1.000 | 0.001 |
| Education | Senior high school | Junior high school or lower | 0.222 | 0.237 | Master or above | Junior high school or lower | 0.695 | 0.152 |
| | | Junior college or undergraduate | 0.720 | 0.060 | | Senior high school | 0.010 | 0.177 |
| | | Master or above | 0.811 | 0.067 | | Junior college or undergraduate | 0.009 | 0.129 |
| Health status | Poor | Normal | 0.275 | 0.288 | Poor | Normal | 0.749 | 0.197 |
| | | Good | <0.001 | 0.769 | | Good | 0.918 | 0.157 |
| | | Very good | <0.001 | 1.178 | | Very good | 0.499 | 0.274 |
| Annual income | 50000–100000 CY | Less than 50000 CY | <0.001 | 0.327 | More than 300000 CY | Less than 50000 CY | 0.119 | 0.283 |
| | | 100000–200000 CY | 0.056 | 0.101 | | 50000–100000 CY | 0.063 | 0.304 |
| | | 200000–300000 CY | 0.113 | 0.172 | | 100000–200000 CY | 0.938 | 0.128 |
| | | More than 300000 CY | 0.714 | 0.153 | | 200000–300000 CY | 0.999 | 0.077 |
| Age | 26–30 | -18 | <0.001 | 0.585 | 61- | -18 | 0.473 | 0.349 |
| | | 18–25 | <0.001 | 0.433 | | 18–25 | 0.034 | 0.368 |
| | | 31–40 | 0.006 | 0.160 | | 26–30 | 0.003 | 0.429 |
| | | 41–50 | <0.001 | 0.245 | | 31–40 | 0.042 | 0.343 |
| | | 51–60 | <0.001 | 0.259 | | 41–50 | 0.998 | 0.120 |
| | | 61- | 0.799 | 0.191 | | 51–60 | 1.000 | 0.038 |
| Current occupation | Nurse | Businessman | <0.001 | 0.461 | Teacher | Businessman | 0.482 | 0.212 |
| | | Officer | <0.001 | 0.504 | | Officer | 1.000 | 0.066 |
| | | Teacher | <0.001 | 0.371 | | Police | 1.000 | 0.112 |
| | | Police | <0.001 | 0.617 | | Farmer | 0.015 | 0.350 |
| | | Farmer | <0.001 | 0.479 | | Employee | 0.781 | 0.206 |
| | | Employee | <0.001 | 0.442 | | Doctor | 0.551 | 0.185 |
| | | Doctor | <0.001 | 0.381 | | Nurse | <0.001 | 0.510 |
| | | Medical technician | 0.007 | 0.340 | | Medical technician | 0.199 | 0.317 |
| | | Retired re-employee | 0.001 | 0.303 | | Retired re-employee | 1.000 | 0.020 |
| | | Non-medical student | <0.001 | 0.740 | | Non-medical student | 0.008 | 0.305 |
| | | Medical student | <0.001 | 0.409 | | Medical student | 0.953 | 0.168 |

CY, Chinese yuan; SGS-scale: the rhythm of sleep, getting-up, and socializing; EPE-scale: the rhythm of eating, physical practice, and entertainment activities; CDCPD, Chronic diseases comorbid with psychosomatic diseases.

education level of senior high school, annual pre-tax income of 50,000–100,000 Chinese Yuan (CY), or chronic disease comorbid with psychosomatic diseases mostly suffered from disrupted rhythms of life and work. The age group of older than 61, and subjects with education levels of master's degree or above, annual pre-tax income over 300,000 CY, or chronic disease without psychosomatic diseases mostly suffered from disrupted rhythms of entertainment. Especially, nurses showed significantly more severe disruption of SGS rhythm, whereas the elderly reported significant irregularity in EPE rhythm (Table 2).

## Psychological impacts in the participants

There were significant differences in SAS and SDS scores among participants with different genders, ages, marital status, education levels, annual pre-tax incomes, health status, current occupation, and chronic diseases with or without psychosomatic diseases in each of the variables (all $P<0.05$) (Table 4). Specifically, nurses and participants with female gender, divorce or widow status, education levels of junior high school or below, poor health status, and chronic diseases comorbid with psychosomatic diseases mostly suffered from depression and anxiety. Whereas the age group of 26–30 years, and participants with annual pre-tax income of 50,000–100,000 CY mostly suffered from depression, the age group of 31–40 years and participants with annual pre-tax income over 300,000 CY mostly suffered from anxiety. Additionally, SDS scores were significantly higher in medical students than in non-medical students. There were significant differences in the prevalence of depression and anxiety among participants with different sociodemographic backgrounds (all $P<0.05$, S1 and S2 Tables). The overall prevalence of depression and anxiety was 24.33% and 12.64%, respectively, as defined by SDS or SAS scores over 50. Among the different occupational groups, nurses had the highest rates of depression (32.94%) and anxiety (18.98%). Participants with chronic diseases combined with psychosomatic diseases had the highest rates of depression (45.90%) and anxiety (31.45%).

## Correlations between the social rhythm and psychological impacts

Among the 5854 participants, mean scores on SGS-scale and EPE-scale were 20.65±9.01 and 8.12±6.00, respectively, and mean SAS score and mean SDS score were 30.51±7.49 and 41.95 ±11.75, respectively. Spearman correlation analysis found that SGS scale was moderately correlated with depression and anxiety, with the correlation coefficients of 0.550 and 0.544, respectively, (both $P<0.001$, S3 Table). In the multivariate linear regression analysis where the scores of SPS-scale or EPE-scale were taken as dependent variables, the score of SDS and SAS were taken as independent variables, and the age, gender, education, and occupation status were taken as confounding factors, the correlations between the scores of SPS-scale and EPE-scale and the scores of SDS and SAS remained unchanged (all $P<0.05$, Table 5, S1 and S2 Figs). The disruption of sleep, getting-up, and socializing rhythm was positively associated with depression and anxiety ($\beta = 0.355$, $P<0.001$; $\beta = 0.186$, $P<0.001$). The disruption of eating, physical practice, and entertainment rhythm showed opposite association to psychological impact: negatively related to depression and positively related to anxiety ($\beta = -0.049$, $P = 0.036$; $\beta = 0.082$, $P<0.001$) (Table 5).

## Discussion

The COVID-19 pandemic has brought great challenges and altered daily routines. It also undoubtedly contributes to emotional stress, fear, sadness, loneliness, anxiety and depression, and thus is associated with a global increase in psychological impacts [5]. One of the many mechanisms by which COVID-19-related changes in social rhythm impact mental health involves circadian disruption [7]. A previous study showed that the early identification and intervention of rhythm disruption are of important clinical significance [21]. Rhythm disruption is part of the pathogenic cascade and is associated with mood disorders, such as anxiety, depression, low happiness, strong loneliness, and bipolar disorder during the non-outbreak period of infectious disease [7, 15, 22]. Limited studies have found that sleep disruption is related to depression in the general population during the COVID-19 pandemic [11]. The present study found that the disruption of social rhythm was quite different among people with various sociodemographic backgrounds. The female gender, low-degree education level,

**Table 4. Scores of depression and anxiety in participants with different sociodemographic backgrounds during the COVID-19 pandemic (N = 5854).**

| Feature | | SDS | | | | SAS | | | |
|---|---|---|---|---|---|---|---|---|---|
| | | Mean±SD | Z/X2 | P | Cohen's d | Mean±SD | Z/X2 | P | Cohen's d |
| Gender | Female | 43.34±11.66 | -11.870 | <0.001 | 0.290 | 38.80±9.39 | 8.441 | <0.001 | 0.173 |
| | Male | 39.97±11.59 | | | - | 37.19±9.23 | | | - |
| Age | -18 | 39.34±10.93 | 63.393 | <0.001 | - | 35.09±8.35 | 266.138 | <0.001 | - |
| | 18–25 | 40.13±11.63 | | | 0.070 | 35.82±8.27 | | | 0.088 |
| | 26–30 | 43.87±12.20 | | | 0.391 | 39.66±9.97 | | | 0.497 |
| | 31–40 | 43.59±11.68 | | | 0.376 | 39.84±9.76 | | | 0.523 |
| | 41–50 | 42.48±11.89 | | | 0.275 | 39.40±9.94 | | | 0.470 |
| | 51–60 | 41.27±10.76 | | | 0.178 | 38.12±8.61 | | | 0.357 |
| | 61- | 41.03±10.97 | | | 0.154 | 38.91±9.32 | | | 0.432 |
| Marital status | Unmarried | 40.65±11.79 | 63.778 | <0.001 | - | 36.26±8.56 | 222.618 | <0.001 | - |
| | Married | 42.78±11.60 | | | 0.182 | 39.39±9.61 | | | 0.344 |
| | Divorced /widowed | 44.57±12.05 | | | 0.329 | 40.87±10.63 | | | 0.478 |
| Education | Junior high school or lower | 45.22±13.29 | 12.821 | <0.05 | 0.281 | 39.85±10.43 | 10.900 | <0.001 | 0.196 |
| | Senior high school | 42.83±11.37 | | | 0.097 | 38.57±9.38 | | | 0.070 |
| | Junior college or undergraduate | 41.71±11.67 | | | - | 37.92±9.24 | | | - |
| | Master or above | 42.31±12.17 | | | 0.050 | 38.86±9.85 | | | 0.098 |
| Annual income | Less than 50000 CY | 40.73±11.74 | 79.210 | <0.001 | 0.107 | 36.49±8.86 | 204.895 | <0.001 | - |
| | 50000–100000 CY | 43.54±11.65 | | | 0.365 | 39.72±9.45 | | | 0.353 |
| | 100000–200000 CY | 42.46±11.79 | | | 0.264 | 39.17±9.67 | | | 0.289 |
| | 200000–300000 CY | 39.56±10.12 | | | - | 36.93±8.02 | | | 0.052 |
| | More than 300000 CY | 42.19±12.56 | | | 0.231 | 40.05±11.42 | | | 0.348 |
| Health status | Poor | 62.90±11.77 | 778.373 | <0.001 | 2.342 | 55.94±12.3 | 827.454 | <0.001 | 2.112 |
| | Normal | 49.73±12.34 | | | 1.149 | 44.56±10.55 | | | 1.120 |
| | Good | 42.00±10.57 | | | 0.526 | 37.95±8.48 | | | 0.466 |
| | Very good | 36.36±10.88 | | | - | 34.14±7.86 | | | - |
| Current occupation | Businessman | 40.42±10.59 | 249.001 | <0.001 | 0.187 | 37.29±8.37 | 374.507 | <0.001 | 0.355 |
| | Officer | 40.25±9.25 | | | 0.182 | 37.86±7.63 | | | 0.449 |
| | Teacher | 41.58±11.24 | | | 0.287 | 38.58±9.38 | | | 0.485 |
| | Police | 38.89±11.34 | | | 0.042 | 36.71±8.89 | | | 0.271 |
| | Farmer | 43.02±11.59 | | | 0.411 | 37.97±9.14 | | | 0.419 |
| | Employee | 42.08±10.68 | | | 0.341 | 38.25±8.34 | | | 0.478 |
| | Doctor | 43.85±12.39 | | | 0.468 | 40.12±10.32 | | | 0.628 |
| | Nurse | 45.17±12.35 | | | 0.583 | 40.41±10.14 | | | 0.668 |
| | Medical technician | 44.74±11.84 | | | 0.558 | 40.14±9.77 | | | 0.653 |
| | Retired re-employee | 42.24±10.47 | | | 0.359 | 38.98±8.61 | | | 0.561 |
| | Non-medical student | 38.43±10.73 | | | - | 34.50±7.32 | | | - |
| | Medical student | 41.93±12.18 | | | 0.305 | 37.93±9.59 | | | 0.402 |
| Chronic disease | No chronic disease | 40.27±11.12 | 363.830 | <0.001 | - | 36.48±8.40 | 36.616 | <0.05 | - |
| | CDCPD | 48.86±12.70 | | | 0.720 | 44.88±10.97 | | | 0.860 |
| | Chronic diseases only | 45.27±11.42 | | | 0.444 | 41.47±9.45 | | | 0.558 |

SD, standard deviation; CY, Chinese yuan; SDS, Zung's self-rating depression scale; SAS, Zung's self-rating anxiety scale; CDCPD, Chronic diseases comorbid with psychosomatic diseases.

lower or higher than average income, poor health status, age group of 26–30 years and older than 61 years, nurses and subjects with divorce mostly suffered from disruption of social rhythm. In addition, the present study observed that the participants with female gender, age

**Table 5. Correlations between the social rhythm disruption and psychological impacts as determined by multivariate linear regressive analysis.**

| | | | Standardized coefficient (β) | t | P | EXP(B) 95% CI | |
|---|---|---|---|---|---|---|---|
| | | | | | | Lower | Upper |
| SGS-scale | Model 1 | SDS | 0.381 | 20.080 | <0.001 | 0.264 | 0.321 |
| | | SAS | 0.203 | 10.689 | <0.001 | 0.160 | 0.231 |
| | Model 2 | SDS | 0.373 | 19.463 | <0.001 | 0.258 | 0.315 |
| | | SAS | 0.200 | 10.385 | <0.001 | 0.156 | 0.229 |
| | Model 3 | SDS | 0.355 | 18.370 | <0.001 | 0.243 | 0.302 |
| | | SAS | 0.188 | 9.791 | <0.001 | 0.145 | 0.217 |
| | Model 4 | SDS | 0.355 | 18.373 | <0.001 | 0.244 | 0.302 |
| | | SAS | 0.186 | 9.645 | <0.001 | 0.143 | 0.216 |
| EPE-scale | Model 1 | SDS | -0.082 | -3.565 | <0.001 | -0.065 | -0.019 |
| | | SAS | 0.109 | 4.780 | <0.001 | 0.041 | 0.099 |
| | Model 2 | SDS | -0.049 | -2.129 | 0.033 | -0.048 | -0.002 |
| | | SAS | 0.082 | 3.582 | <0.001 | 0.024 | 0.082 |
| | Model 3 | SDS | -0.049 | -2.101 | 0.036 | -0.048 | -0.002 |
| | | SAS | 0.081 | 3.513 | <0.001 | 0.023 | 0.081 |
| | Model 4 | SDS | -0.049 | -2.101 | 0.036 | -0.048 | -0.002 |
| | | SAS | 0.082 | 3.518 | <0.001 | 0.023 | 0.081 |

SGS-scale: the rhythm of sleep, getting-up, and socializing; EPE-scale: the rhythm of eating, physical practice, and entertainment activities; SDS/ SAS: Zung's self-rating depression and anxiety scales.

Model 1: uncorrected; Model 2: corrective factors: gender, age, marital status, and education level; Model 3: corrective factors: based on the corrective factors of Model 2 and health status, annual pre-tax income, and current occupation; and Model 4: corrective factors: based on the corrective factors of Model 3 and psychosomatic diseases. The results of Model 4 were presented in this table

group of 26–40 years, divorce or widow status, low education level, lower or higher than average income, poor health status, and chronic diseases comorbid with psychosomatic diseases, and nurses mostly suffered from depression and/or anxiety.

There are two major possible explanations for the observed gender differences in rhythm disruption, depression, and anxiety. First, the steroid receptors are expressed in almost every site that receives direct suprachiasmatic nucleus (SCN) input [23]. The endogenous circadian clocks can be reset by estrogen hormone signals in women, affecting the balance of the hypothalamus-pituitary-adrenal (HPA) axis, and making the HPA axis more responsive to stress than males [23, 24]. The disruption of circadian rhythms within the HPA axis, and the sleep-arousal system differs between the genders, and is associated with dysfunction and disease [23]. Understanding gender differences in the circadian timing system can lead to improved treatment strategies. Second, the psychosocial contexts differ between the genders, and the impact of traditional gender roles and socio-cultural influences on stress and anxiety are greater in women than in men [25].

In this study, we found that people with low education levels or low income mostly suffer from irregular social rhythms, likely due to their weak competitiveness in the job market and inability to adapt to the new E-commerce models during the COVID-19 pandemic due to the isolation or even lockdown [26]. Moreover, they did not have sufficient judgment while facing various, overloaded and timely information on the COVID-19 pandemic, and easily felt stressed, depressive and anxious [27]. Interestingly, both the people with high incomes and those with poor health status were mostly suffered from disrupted rhythms. On one hand, the disrupted rhythm in people with high incomes may be explained as follows: during the COVID-19 pandemic, the consumption expenditures of high-income groups dropped sharply

by as much as 17%, while the expenditures of low-income families dropped by only 4% [28]. At the same time, entertainment venues that provide services for high-income groups, such as cultural activity centers, clubs and gymnasiums, golf playgrounds, luxury party venues, were mostly closed. All these changes would impact severely on the social rhythms of these groups of people. On the other hand, the disrupted rhythm in people with poor health status can be explained that the regular activities, such as walking, jogging, swimming and fitness, which were significantly restricted since their poor health status with basic diseases, weak immune function, and low ability to fight pathogens or adapt to a rhythm change [29]. In addition, our finding that a high frequency of the irregular rhythm in the divorced group might be related to the disintegration of the family, loose family relations, economic problems, and the problem of raising children as a single parent. Especially, the isolation during the COVID-19 pandemic might have aggravated the fear for the unknown future, loneliness, helplessness, or irregular lifestyle [30].

The present study found that the people aged 26–40 years old and the elderly reported similarly high SGS-scale scores and were vulnerable to circadian disruption. The former is the main force of the society that promote the consumption, cultural creativity, and economic development [31]. However, when they must stay at home or fight against the COVID-19 pandemic due to the sudden outbreak of the disease, it is difficult for them to maintain a normal social rhythm. It is worth noting that the elderly reported the highest rate of rhythm disruption of eating, physical practice, and entertainment behaviors in the present study. Previously, Liu *et al*. [32] conducted a study on lifestyle and the health status of Chinese elderly, which included the elderly who lived in their own houses, with 20.6% of them being widows or widowers. They found that lack of physical practice, advanced age, and alcohol consumption were risk factors for their self-rate health status whereas a long sleep time was the protective factor [33]. Indeed, it has been reported that advanced age is characterized by a progressive decreasing amplitude and phase advance of circadian rhythmicity in overall biological functions, including blood pressure, hormonal circadian secretions, and immune function changes [33]. The elderly could not go out for dancing after dinner, exercise in the morning, or visit old friends or relatives due to recommended social isolation during the COVID-19 pandemic. Furthermore, they were inherently at high risk for COVID-19 infection and easily generated negative emotions [34]. These various factors were prone to form a vicious circle.

It is noteworthy to mention that nurses had the highest score of rhythm disruption in going to bed, get-up, and socializing among occupations, suggesting that they were more affected by the pandemic than other occupations such as businessmen, officers, teachers, police, farmers, doctors, and students. Night shift work has been thought to play a role through a disruption of the circadian rhythm, decreased synthesis of melatonin and sleep deprivation [35]. We assume that nurses undertake a working articulation in shifts to keep the continuity of healthcare throughout a whole day, and the COVID-19 pandemic results in heavy workload, insufficient rest, quarantine, and social isolation. Consequently, the already dysfunctional social rhythm of nurses is further exacerbated. McElroy *et al*. reported that night and long shifts caused fatigue and adversely affected the family and social life of hospital employees [36]. Rosa *et al*. indicated that shift work could cause psychological consequences, including anxiety, stress, and depression, which is an obstacle for social and family relationships and a risk factor for metabolic disorders, diabetes, cardiovascular disorders, and breast cancer [37].

The present study firstly reported the critical connection between social rhythm disruption and psychological impacts in a large-scale population under the stress of such a pandemic. Our study further found that the scores of social rhythms were positively correlated with SDS and SAS scores under the stress of the COVID-19 pandemic. The following factors are speculated to be involved in the connection. The first factor is the social zeitgeber theory. The notion

that life events increase the likelihood of mood episodes *via* decreased zeitgeber scaffolding for a vulnerable circadian system has been known as the social zeitgeber hypothesis, and the "social rhythm" refers to the cross-day stability of the timing of behaviors that act as zeitgebers [7]. Social stimuli under the stress of COVID-19 may affect the circadian behavioral programs by regulating the phase and period of circadian clocks (*i.e.* a zeitgeber action, either direct or by conditioning to photic zeitgebers) which leads to various adverse mental health outcomes [38, 39]. In turn, depressive episodes that arise because of life events disturbing social zeitgebers subsequently affect biological rhythms, and thus derail social and biological rhythms [39, 40]. This present study expanded the scope of application of these rhythm theories from patients with mood disorders to the population with different sociodemographic backgrounds. The second factor is the lockdown protocols that have been adopted to minimize COVID-19 pandemic transmission. The protocols may weaken the zeitgeber-setting mechanism, and impact the stability of the timing of daily behaviors, such as sleeping, getting up, socializing, eating, physical practice, and entertainment, which, in turn, lead to mental health disorders, such as anxiety and depression [7]. The third factor is the daily function of HPA activity, which is also regulated by the biological clock [41]. Anxiety disorders and major depressive disorder have been associated with increased and blunted HPA axis reactivity to social stress respectively, which engages the diverse biological pathways that bolster successful stress adaptation and promote stress resilience. Adaptation to the challenges of stress is compromised when these pathways are no longer functioning optimally, resulting in the disrupted rhythms on psychological impacts [42]. Of course, other mechanisms may also connect social rhythm dysregulation with psychological impacts, including circadian gene variations [43] and social jetlag [44].

Based on the findings of the present study, more effective measures that combine circadian rhythms with the diagnosis and treatment of psychosomatic diseases may improve psychological impacts and maximize the effectiveness of treatment. Therefore, the following measures are recommended for stabilizing daily routines, especially for those specific population who mostly suffered from disrupted rhythms: 1), self-management strategies for setting up and increasing regularity of daily activities and lifestyle habits during the COVID-19 pandemic [45]; 2), spending 1 hour/day outdoors, walking for more than one hour a day and basking in the sun, which may regulate the body clock by the light-dark cycle [7]; 3), scheduling social interactions at the same time of the day, and seeking the best candidates who may share thoughts and feelings in real time [7]; 4), eating meals at the same time every day, and adjusting the eating patterns and food categories. 5), avoiding daylight hour naps and the nocturnal blue spectrum light pollution from computers, mobile phones, electronic tablets and televisions, which suppresses the sleep-helping hormones [45]. Chronotherapy seems to be a promising approach to restore the proper circadian pattern in the elderly with an obvious sleep problem through adequate sleep hygiene, timed light exposure, and the use of a chronobiotic medication [12]; 6), taking drugs that adjust the biological clock for nurses with sleep disorders caused by nurses' shift system when necessary [46]; and 7), encouraging online therapy program for normalizing the disrupted social rhythms in COVID-19 environment, with the support of health care professionals and persuasive systems design and user experience/ interaction features [7].

The present study provides new ideas for the simultaneous intervention from the perspective of social rhythms and psychology, and would lay an epidemiological foundation for the interaction of social rhythms with the diagnosis and treatment of psychosomatic diseases and the maximization of therapeutic efficacy. However, there are a few limitations in the present study. First, this cross-sectional study cannot reveal causality, and the nature of voluntary participation may result in selection bias. Second, in this study, we did not track the 24-hour

circadian rhythm. Indeed, it is necessary to track the circadian rhythms/sleep-wake cycle by a multi-lead-sleep monitor considering different psychological impacts, which will be implemented in our further research. Third, the survey response rate was not obtained since this online-survey was participated voluntarily and the lockdown and social isolation policy did not encourage a face-to-face survey, and it was difficult to get the survey response rate. Fourth, we did not consider obtaining the individual informed consent for teenagers themselves under 18 years old, in addition to their parents' consent, which will also be implemented in our further similar research.

In conclusion, there exist rhythm disruptions among Chinese people with different socio-demographic backgrounds, which are closely associated with psychological impacts under the stress of the COVID-19 pandemic. Social rhythm disruption is independently associated with depression and anxiety. Interventions should be applied to the people vulnerable to the rhythm disruptions during the COVID-19 pandemic.

## Supporting information

**S1 Fig.**
(TIF)

**S2 Fig.**
(TIF)

**S1 Dataset.**
(XLSX)

**S1 Table. Prevalence of depression with different grades in participants (N = 5854).**
(DOCX)

**S2 Table. Prevalence of anxiety with different grades in participants (N = 5854).**
(DOCX)

**S3 Table. Spearman's correlation between subscale 1, subscale 2, Zung's self-rating depression scale (SDS) and Zung's self-rating anxiety scale (SAS).** (N = 5854).
(DOCX)

## Acknowledgments

Zhihua Gao, PhD and Jinxia Zhu, PhD, provided valuable assistance with on questionnaire design. Kai Guo provided the on-line technical support. We are grateful for Medjaden Inc. for its assistance in the preparation of the manuscript and professional editing and language polishing.

## Author Contributions

**Data curation:** Ping He, Jing Wang, Yanjun Wang, Hui Lin.

**Formal analysis:** Hui Lin.

**Funding acquisition:** Wo Wang, Li Kuang.

**Project administration:** Bin Wang, Mei Zhao, Mingming Deng, Dongfeng Chen.

**Supervision:** Tianwei Deng.

**Writing – original draft:** Min Yang.

**Writing – review & editing:** Xiaoming Xu, Dan Li.

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
