## [Decision Letter · Decision Letter 0]

6 Nov 2020

PONE-D-20-26146

Disrupted Rhythms of Life, Work and Entertainment Behaviors and Their Associations with Mental Health Problems under the Stress of COVID-19 Epidemic: A Survey in 5854 Chinese People with Different Backgrounds

PLOS ONE

Dear Dr. Chen,

Thank you for submitting your manuscript to PLOS ONE. After careful consideration, we feel that it has merit but does not fully meet PLOS ONE’s publication criteria as it currently stands. Therefore, we invite you to submit a revised version of the manuscript that addresses the points raised during the review process.

Please take note of the reviewers' comments at the bottom of this email as well as my own comments under Additional Editor Comments below.

We look forward to receiving your revised manuscript.

Kind regards,

Ben Bullock

Academic Editor

PLOS ONE

Journal Requirements:

3. Please include additional information regarding the survey or questionnaire used in the study and ensure that you have provided sufficient details that others could replicate the analyses.

For instance, if you developed a questionnaire as part of this study and it is not under a copyright more restrictive than CC-BY, please include a copy, in both the original language and English, as Supporting Information. 

If the original language is written in non-Latin characters, for example Amharic, Chinese, or Korean, please use a file format that ensures these characters are visible.

4. In your Methods section, please provide additional information about the participant recruitment method and the demographic details of your participants. Please ensure you have provided sufficient details to replicate the analyses such as:   

-    a description of any inclusion/exclusion criteria that were applied to participant recruitment,

-    a statement as to whether your sample can be considered representative of a larger population,

-    a description of how participants were recruited, and

-       descriptions of where participants were recruited and where the research took place."

5. In your Methods section, please include additional information about your dataset and ensure that you have included a statement specifying whether the collection method complied with the terms and conditions for the website."

6. Thank you for stating the following at the end of your manuscript:

'Funding/Support: This study was supported by the grant from Chongqing basic research and foreword exploration project (Grant No. CSTC2018jcyjAX0620).

Role of the Funder/Sponsor: The sponsors had no role in the design and conduct of the study; collection, management, analysis, and interpretation of the data; preparation, review, or approval of the manuscript; and decision to submit the manuscript for publication.

'Include this sentence at the end of your statement: The funders had no role in study design, data collection and analysis, decision to publish, or preparation of the manuscript.'

7. Please amend the manuscript submission data (via Edit Submission) to include author Hui Lin.

Additional Editor Comments:

Thank you for your interesting and timely research manuscript. The topic is clearly of current interest, however major revisions are necessary in order for this manuscript to be publishable. Reviewer 2 in particular provides comprehensive feedback on areas of the manuscript requiring significant revision.

Both reviewers recommend that the manuscript would benefit from thorough editing of language usage, spelling, and grammar by a fluent or native speaker of English. If you do not know anyone who can help you do this, you may wish to consider employing a professional scientific editing service.

In addition to the feedback provided by the reviewers I suggest that any revised submission particularly focus on improving the Discussion. A more sophisticated discussion around HPA axis activity, SCN regulation of circadian rhythms, and social rhythms is necessary (see also Reviewer 2's feedback on the Discussion). For example, the relevance of the "knockout monkey" model is unclear. I also suggest a more sophisticated discussion of the possible explanations for the observed gender differences in stress and anxiety. The manuscript relies on evidence from rat studies and, unusually, transgender individuals, to support increased HPA axis activity in females. A more nuanced discussion is needed that may or may not include reference to these biological differences between men and women, but that also considers differences in psychosocial contexts between the sexes and the impact of traditional gender roles and socio-cultural influences on higher rates of stress and anxiety among women in comparison to men. There is also an unusual reference to the impact of menopause on "women's circadian and endocrine disorders" that is not supported by a reference, nor is it a measured variable in the study.

Reviewers' comments:

Reviewer's Responses to Questions

**Comments to the Author**

1. Is the manuscript technically sound, and do the data support the conclusions?

Reviewer #1: Yes

Reviewer #2: Partly

2. Has the statistical analysis been performed appropriately and rigorously? 

Reviewer #1: No

Reviewer #2: Yes

3. Have the authors made all data underlying the findings in their manuscript fully available?

Reviewer #1: Yes

Reviewer #2: Yes

4. Is the manuscript presented in an intelligible fashion and written in standard English?

Reviewer #1: No

Reviewer #2: Yes

5. Review Comments to the Author

Reviewer #1: This is a timely and novel research project.

Overall language - While there is a very good flow of idea and structure to the manuscript, the language is awkward and redundant at times, and would benefit from additional editing by a fluent or native speaker of English.

Statistical analysis - Please considering re-calculating probabilities while correcting for multiple comparisons/tests. Please consider reducing the table content to improve readability.

Reviewer #2: 1. Is the manuscript technically sound, and do the data support the conclusions? See comments on the discussion

2. Has the statistical analysis been performed appropriately and rigorously? No revisions

3. Have the authors made all data underlying the findings in their manuscript fully available? No revisions

4. Is the manuscript written in an intelligible fashion and written in standard English?

While I answered yes, I think this manuscript calls for improvement and revisions in this category before publication. For example in the abstract the background and objective are awkward and hard to follow. As is, they read: “The rhythms of life, work and entertainment behaviors are considered as the external behavioral manifestations of biological rhythm. The present study aimed to evaluate the distinctive disrupted rhythms of behaviors and their associations with mental health problems in people with different backgrounds under the stress of COVID19 epidemic.” Social rhythms such as work and entertainment routines are driven by biological rhythms including the sleep wake cycle. The present study aimed to evaluate if disruption of social rhythms and routines were associated with mental health problems under the stress of the COVID19 epidemic and if this differed by people of XXX.”

Additional comments:

Introduction:

1. The Introduction needs to be expanded and to include more of the relevant literature. You seem to be missing a large body of research on this topic. Please see the article by Murray, Gottleib, & Swartz (2020) Maintaining Daily Routines to Stabilize Mood: Theory, Data, and Potential Intervention for Circadian Consequences of COVID-19. Canadian Journal of Psychiatry. If you look at the reference list of this paper, you will find more literature on this topic not included in your Background section.

2. Specify what groups you are comparing in your aims.

3. Is this exploratory or do you have hypotheses?

Methods:

1. In the limitations I would include that those under 18, at least those over 13 could have signed/checked their own assent in addition to their parents’ consent.

2. I suggest giving brief information on the psychometric properties for your instruments.

3. Did you track ask about circadian rhythms/sleep wake cycle? If not, this is a limitation

Statistical Analysis:

Specify what groups you are comparing. If a one-way t-test was used specify your hypothesis after your aims.

Results:

1. What was the response rate?

2. Give an example of why some questionnaires were invalid and therefore excluded.

3. In section 4 (Correlation analysis) specify what you mean by low score and high score group the first time you say it. Also, specify what the subscales are instead of referring to them as subscale1, etc.

Discussion:

There seems to be a lot of speculation in this section and assumptions that is not backed up by research, the literature or the data. You may want to back up your paper/assertions with theory. See: Social Rhythms Disruption Theory.

6. PLOS authors have the option to publish the peer review history of their article (what does this mean?). If published, this will include your full peer review and any attached files.

Reviewer #1: No

Reviewer #2: No

---

## [Author Response · Author response to Decision Letter 0]

25 Jan 2021

Dear Dr. Bullock,

Thank you for your decision letter and the reviewers for their time in reviewing our manuscript ( ID: PONE-D-20-26146), entitled “Disrupted rhythms of life, work and entertainment and their associations with psychological impacts under the stress of the COVID-19 pandemic: A Survey in 5854 Chinese people with different sociodemographic backgrounds”. According to your insightful advice and the thoughtful comments and suggestions of the reviewers, we have significantly revised our manuscript. All changes are highlighted in red in the revised manuscript. In addition, we have responded to the reviewers’ comments in a point-by-point format, which are listed below this letter. We would like to re-submit it for your consideration. We hope that the revision is acceptable, and we look forward to hearing from you soon.

Response to the comments of Reviewer #1: 

1. This is a timely and novel research project.

Overall language - While there is a very good flow of idea and structure to the manuscript, the language is awkward and redundant at times, and would benefit from additional editing by a fluent or native speaker of English.

Response: Thanks for your positive and encouraging comment on our research project. According to your suggestion, we have sent our manuscript to Medjaden Inc, a professional editing and proofreading service, for language polishing. 

2. Statistical analysis - Please considering re-calculating probabilities while correcting for multiple comparisons/tests. 

Response: Thanks for your professional suggestion. Accordingly, we have re-calculated probabilities while correcting for multiple comparisons. For more clarity and readability, we have split the original Table 1 into four tables, i.e. Table 1 (General sociodemographic characteristics of 5854 analyzed participants) (Page 9, Line 15 — Page 10, Line 1); Table 2 (Scores of social rhythms in participants with different sociodemographic backgrounds during the COVID-19 pandemic (N=5854) (Page 11, Line 6 — Page 12, Line 3); Table 3 (Comparison of SGS-scale and EPE-scale scores between the group with the highest score and each of other groups in terms of sociodemographic backgrounds) (Page 12, Line 4 — Page 14, Line 3); Table 4 (Score of depression and anxiety in participants with different sociodemographic backgrounds during the COVID-19 pandemic, N=5854) (Page 15, Line 4 — Page 16, Line 3). The overall �2, Z and P-values in the multiple comparisons are shown in Table 2. Additionally, we have conducted comparisons between two groups for each variable and presented the results for the comparisons between the group with the highest score and each of other groups in SGS-scale (Table 3) (Page 12, Line 4 — Page 14, Line 3).

3. Please consider reducing the table content to improve readability.

Response: As described above, we have split the original Table 1 into four tables (Tables 1-4) for clarity, and deleted the columns of SAS and SDS in Table 1 in the revised manuscript. We have labeled the original Tables 2, 3 and 4 as Supplementary Tables 1 and 2, and 3 in the revised manuscript (Page 38-40). In addition, we have combined the original Tables 5 and 6 into a single table (Table 5) in the revised manuscript (Page 17, Line 10—Page 18, Line 3). Thus, the formal table number has been reduced from 6 to 5, with three complementary tables in the revised manuscript. In addition, we have labeled the original Figures 1 and 2 as Supplementary Figures 1A &B in the revised manuscript (Page 37).

Responses to the comments of Reviewer #2:

1. Is the manuscript technically sound, and do the data support the conclusions? See comments on the discussion

Response: Thanks for your positive and valuable comments. Our responses are listed below your comments on the Discussion accordingly. Moreover, we have significantly revised the Discussion section of the revised manuscript (Page 19, Lines 11-20; Page 21, Lines 6-21; Page 22, Lines 1- 21; Page 23, Lines 1-21 and Page 24, Lines 1-4).

2. Has the statistical analysis been performed appropriately and rigorously? No revisions

Response: Thanks for your professional suggestion. Accordingly, we have re-calculated probabilities while correcting for multiple comparisons. For more clarity and readability, we have split the original Table 1 into four tables, i.e. Table 1 (General sociodemographic characteristics of 5854 analyzed participants) (Page 9, Line 15 — Page 10, Line 1); Table 2 (Scores of social rhythms in participants with different sociodemographic backgrounds during the COVID-19 pandemic (N=5854) (Page 11, Line 6 — Page 12, Line 3); Table 3 (Comparison of SGS-scale and EPE-scale scores between the group with the highest score and each of other groups in terms of sociodemographic backgrounds) (Page 12, Line 4 — Page 14, Line 3); Table 4 (Score of depression and anxiety in participants with different sociodemographic backgrounds during the COVID-19 pandemic, N=5854) (Page 15, Line 4 — Page 16, Line 3). The overall �2, Z and P-values in the multiple comparisons are shown in Table 2. Additionally, we have conducted comparisons between two groups for each variable and presented the results for the comparisons between the group with the highest score and each of other groups in SGS-scale (Table 3) (Page 12, Line 4 — Page 14, Line 3).

3. Have the authors made all data underlying the findings in their manuscript fully available? No revisions.

Response: Thanks for your professional suggestion. We have made all data underlying the findings in the revised manuscript fully available. According to your insightful advice, these data have been reflected in the revised version of the Results, Discussions, and Supplementary materials.

4. Is the manuscript written in an intelligible fashion and written in standard English?

While I answered yes, I think this manuscript calls for improvement and revisions in this category before publication. For example in the abstract the background and objective are awkward and hard to follow. As is, they read: “The rhythms of life, work and entertainment behaviors are considered as the external behavioral manifestations of biological rhythm. The present study aimed to evaluate the distinctive disrupted rhythms of behaviors and their associations with mental health problems in people with different backgrounds under the stress of COVID19 pandemic.” Social rhythms such as work and entertainment routines are driven by biological rhythms including the sleep wake cycle. The present study aimed to evaluate if disruption of social rhythms and routines were associated with mental health problems under the stress of the COVID19 pandemic and if this differed by people of XXX.”

Response: Thanks for your good suggestion. As for manuscript writing and language, we have tried our best to revise and improve the manuscript. In addition, we have sent our revised manuscript to Medjaden Inc, a professional editing and proofreading service, for language polishing.

Additional comments:

Introduction:

1. The Introduction needs to be expanded and to include more of the relevant literature. You seem to be missing a large body of research on this topic. Please see the article by Murray, Gottleib, & Swartz (2020) Maintaining Daily Routines to Stabilize Mood: Theory, Data, and Potential Intervention for Circadian Consequences of COVID-19. Canadian Journal of Psychiatry. If you look at the reference list of this paper, you will find more literature on this topic not included in your Background section.

Response: Thanks for your professional suggestion, and providing the valuable article, which we have learned and cited in the Introduction and Discussion sections of the revised manuscript (Page 5, Lines 13-14; Page 22, Lines 14-21; Page 23, Lines 1-9; Page 23, Lines 9-21 and Page 24, Lines 1-4). Accordingly, we have made revisions with more literature on this topic in the Introduction section of the revised manuscript (Reference 5, 7, 23, and 24 ). 

2. Specify what groups you are comparing in your aims.

Response: Thanks for your professional suggestion. The aim of the present study was to evaluate the social rhythm and its associations with the psychological impacts during the initial phase of the COVID-19 pandemic among Chinese people with different sociodemographic backgrounds including gender, age, marital status, education level, annual income, health status, current occupation, and chronic disease status. Moreover, we also compared the group with the highest SGS-scale and EPE-scale scores with each of other groups in terms of the above-mentioned variables (Table 3) (Page 12, Line 4 — Page 14, Line 3). Special groups referred to nurses, the elderly and subjects being divorced or with chronic diseases or psychosomatic diseases who were mostly suffered from disrupted rhythms of life-work behaviors. This point has been clarified in the revised manuscript (Page 10, Lines 7-14; Page 11, Lines 1-5).

3. Is this exploratory or do you have hypotheses?

Response: Yes, we have a hypothesis. We hypothesize that people with different sociodemographic backgrounds have different degrees of rhythm disruption in life, work, and entertainment, that the rhythm disruption is more severe in some specific groups, and that the rhythm disruption is associated with psychological impacts during the initial phase of the COVID-19 pandemic. Accordingly, we have stated the hypotheses clearly in the Introduction of the revised manuscript (Page 6, Lines 4-8).

Methods:

1. In the limitations I would include that those under 18, at least those over 13 could have signed/checked their own assent in addition to their parents’ consent.

Response: Thanks for your thoughtful suggestion. Indeed, we did not consider obtaining the individual informed consent for teenagers themselves, in addition to their parents’ consent. According to your suggestion, we have added it as a limitation in the Discussion section of the revised manuscript (Page 25, Line 21 and Page 26 Lines 1-2). 

2. I suggest giving brief information on the psychometric properties for your instruments.

Response: Thanks for your professional suggestion. Accordingly, we have briefly described the psychometric properties for the instruments used in our study with references to the initial version and the Chinese version of SAS and SDS to interpret their reliability and validity in the revised manuscript (Zung’s Depression and Anxiety Scale, Page 8, Lines 11-14).

3. Did you track ask about circadian rhythms/sleep wake cycle? If not, this is a limitation.

Response: Thanks for your thoughtful suggestion. In this study, we did not track the 24-hour circadian rhythm. Indeed, it is necessary to track the circadian rhythms/sleep wake cycle by multilead-sleep monitor considering different psychological impacts, which will be implemented in our further research. According to your suggestion, we have added this point as a limitation in the Discussion section of the revised manuscript (Page 25, Lines 15-18).

Statistical Analysis:

Specify what groups you are comparing. If a one-way t-test was used specify your hypothesis after your aims. 

Response: Thanks for your professional suggestion. In the present study, SGS-scale and EPE-scale scores reflecting the life-work and entertainment activities, respectively were compared among Chinese people with different sociodemographic backgrounds including gender, age, marital status, education level, annual income, health status, current occupation, and chronic disease status. Moreover, the group with the highest score was compared with each of other groups in terms of above-mentioned variables by using one-way ANOVA with a post hoc test (Table 3) (Page 12, Line 4 — Page 14, Line 3). This point has been clearly stated in the Statistical analysis of the revised manuscript (Page 8, Lines 18-19). In addition, we have specified our hypothesis after the aim (Page 6, Lines 4-8).

Results:

1. What was the response rate?

Response: Thanks for your pertinent question. It was challenging to get the survey response rate because the lockdown and social isolation policy did not encourage a face-to-face survey. Therefore, we conducted the present study using online-survey through a public media (i.e. EQxiu online questionnaire platform), to which participation was voluntary. Accordingly, we have added this point as a limitation in the Discussion section of the revised manuscript (Page 25, Lines 18-21). 

2. Give an example of why some questionnaires were invalid and therefore excluded.

Response: Eighteen questionnaires with invalid were excluded the data analysis due to frivolous or incomplete answers, or the same option for different questions or orderly answers. The information on the invalid questionnaires have been clearly presented in the Results section of the revised manuscript (Page 9, Lines 10-12).

3. In section 4 (Correlation analysis) specify what you mean by low score and high score group the first time you say it. Also, specify what the subscales are instead of referring to them as subscale1, etc.

Response: Thanks for carefully reviewing the manuscript and raising the suggestions. We have deleted the classification of “low score and high score groups”. For more readability, we have renamed the two subscales of the social rhythm scale (i.e. subscale1 and subscale2) as SGS-scale and EPE-scale, respectively, in the revised manuscript. Specifically, the first eight items constituted SGS-scale, which addresses sleeping, getting up, and socializing rhythm, and the last nine items constituted the EPE-scale, which addresses eating, physical practice, and entertainment rhythm. These terms have been clearly defined in the Measures section of the revised manuscript (Page 7, Lines 16-20).

Discussion:

There seems to be a lot of speculation in this section and assumptions that is not backed up by research, the literature or the data. You may want to back up your paper/assertions with theory. See: Social Rhythms Disruption Theory.

Response: Thanks for your critical and constructive comment and suggestion. Accordingly, we have performed literature search and learned several theories on social rhythm, such as Social Zeitgeber Theory. Accordingly, we have extensively revised the Discussion section of the revised manuscript (Page 22, Lines 13-21; Page 23, Lines 1-21; Page 24, Lines 1-4). 

There are five parts (paragraphs) in the Discussion section of the revised manuscript, including: 1), the characteristic analysis of disrupted rhythms with different sociodemographic backgrounds under the stress of COVID-19 pandemic, especially susceptible population suffering from rhythm disruption; 2), the correlations between rhythm disruption and psychological impacts, with explanations of disrupted rhythms on psychological impacts; 3), strategic measurement to restore the rhythm pattern; and 4), limitation; and 5), conclusion.

Responses to Additional Editor Comments:

1.Thank you for your interesting and timely research manuscript. The topic is clearly of current interest, however major revisions are necessary in order for this manuscript to be publishable. Reviewer 2 in particular provides comprehensive feedback on areas of the manuscript requiring significant revision.

Response: We have revised this manuscript according to the two reviewers’ comments, with point-by-point responses.

2. Both reviewers recommend that the manuscript would benefit from thorough editing of language usage, spelling, and grammar by a fluent or native speaker of English. If you do not know anyone who can help you do this, you may wish to consider employing a professional scientific editing service.

Response: Thank you for the advice. We have tried our best to improve the manuscript. In addition, we have sent our revised manuscript to Medjaden Inc, a professional editing and proofreading service, for language polishing.

3.In addition to the feedback provided by the reviewers I suggest that any revised submission particularly focus on improving the Discussion. A more sophisticated discussion around HPA axis activity, SCN regulation of circadian rhythms, and social rhythms is necessary (see also Reviewer 2's feedback on the Discussion). For example, the relevance of the "knockout monkey" model is unclear. I also suggest a more sophisticated discussion of the possible explanations for the observed gender differences in stress and anxiety. The manuscript relies on evidence from rat studies and, unusually, transgender individuals, to support increased HPA axis activity in females. A more nuanced discussion is needed that may or may not include reference to these biological differences between men and women, but that also considers differences in psychosocial contexts between the sexes and the impact of traditional gender roles and socio-cultural influences on higher rates of stress and anxiety among women in comparison to men. There is also an unusual reference to the impact of menopause on "women's circadian and endocrine disorders" that is not supported by a reference, nor is it a measured variable in the study.

Response: Thanks for your insightful and constructive advice. We have performed literature search, read relevant articles and revised Discussion section of the revised manuscript according to your professional suggestion and the advice of reviewer 2 (Page 19, Lines 11-20; Page 20, Lines 1-2; Page 21, Lines 6-21; Page 22, Lines 1-21; Page 23, Lines 1-21; Page 24, Lines 1-4 ; Page 25, Lines 5-8 and Lines 15-21; Page 26, lines 1-2.

---

## [Decision Letter · Decision Letter 1]

16 Mar 2021

PONE-D-20-26146R1

Disrupted rhythms of life, work and entertainment and their associations with psychological impacts under the stress of the COVID-19 pandemic: A Survey in 5854 Chinese people with different sociodemographic backgrounds

PLOS ONE

Dear Dr. Chen,

Thank you for submitting your manuscript to PLOS ONE. After careful consideration, we feel that it has merit but does not fully meet PLOS ONE’s publication criteria as it currently stands. Therefore, we invite you to submit a revised version of the manuscript that addresses the points raised during the review process.

Dear Authors,

Thank you for addressing the reviewer's feedback and submitting a vastly improved manuscript. I have decided to proceed to a decision on the basis of only one reviewer plus my own thorough reading of the manuscript. Reviewer 2 supports acceptance of the manuscript, however I feel some minor amendments are necessary to ensure absolute clarity of meaning in the article. I have added my comments and amendments directly to the revised manuscript (see the attached document). As well as these comments and amendments please ensure the reference list at the end of the manuscript is presented in a consistent style.

We look forward to receiving your revised manuscript.

Kind regards,

Ben Bullock

Academic Editor

PLOS ONE

Journal Requirements:

Additional Editor Comments (if provided):

Please submit you revised manuscript after addressing the Editor comments in the attached document.

Reviewers' comments:

Reviewer's Responses to Questions

**Comments to the Author**

1. If the authors have adequately addressed your comments raised in a previous round of review and you feel that this manuscript is now acceptable for publication, you may indicate that here to bypass the “Comments to the Author” section, enter your conflict of interest statement in the “Confidential to Editor” section, and submit your "Accept" recommendation.

Reviewer #2: All comments have been addressed

2. Is the manuscript technically sound, and do the data support the conclusions?

Reviewer #2: Yes

3. Has the statistical analysis been performed appropriately and rigorously? 

Reviewer #2: Yes

4. Have the authors made all data underlying the findings in their manuscript fully available?

Reviewer #2: Yes

5. Is the manuscript presented in an intelligible fashion and written in standard English?

Reviewer #2: Yes

6. Review Comments to the Author

Reviewer #2: (No Response)

7. PLOS authors have the option to publish the peer review history of their article (what does this mean?). If published, this will include your full peer review and any attached files.

Reviewer #2: No

---

## [Author Response · Author response to Decision Letter 1]

9 Apr 2021

Dear Dr. Bullock,

Thank you very much for your encouraging decision and helpful comments on the revised version of our manuscript, titled “PONE-D-20-26146R1: Disrupted rhythms of life, work and entertainment and their associations with psychological impacts under the stress of the COVID-19 pandemic: A Survey in 5854 Chinese people with different sociodemographic backgrounds”. We also thank the reviewer for endorsing the publication of our manuscript in PLOS ONE. We have further revised the manuscript, and highlighted all changes in the version with Track Changes. In addition, our responses to your comments, in a point-by-point format, are listed below:

Abstract-Method: 

1. Add the add the word "also" in “correlations of SDS and SAS with SGS-scale and EPE-scale were analyzed.”

Response: We have added the word “also” in the sentence in the newly revised manuscript (Page 3, Line 16).

Abstract-Results: 

2. Clarify the mean of “too-low or too-high” in “The scores were significantly higher in the groups with female gender, low education level, too-low or too-high income, poor health status, ages of 26-30 years or older than 61 years, nurses and subjects with divorce or widow status.” 

Response: We accept your suggestion and have changed the “too-low or too-high” into “lower or higher than average” in the newly revised manuscript (Page 3, Line 21).

Introduction

3. delete “(Bashir, M. F. 2020)” to keep referencing style consistent throughout.

Response: We have deleted it from in the newly revised manuscript.

4. Change the word “alarming” in “Simultaneously, the COVID-19 pandemic has alarming implications for a global increase in mental health problems, and the psychological burden during the COVID-19 pandemic has been reported.” to "negative" or "significant" to avoid overstatement.

Response: We have changed it into the word “significant” in the newly revised manuscript (Page 5, Line 10).

5. Add the word “negative” before “psychological impacts” in “COVID-19-related social changes, particularly lockdown protocols, have indeed impacted the timing of daily behaviors in stabilizing circadian function, and social rhythm disruption may be correlated with psychological impacts”.

Response: We have added the word “negative” before psychological impacts in the newly revised manuscript (Page 5, Line 16).

6. Change the word “elder” to “elderly” in “Recently, several studies have reported social and circadian rhythm disruptions in students, office workers, the elder, and patients with narcolepsy or autism.”

Response: We have changed the “elder” to “elderly” in this sentence in the newly revised manuscript (Page 5, Line 20).

7. These hypotheses are not specific enough. Hypotheses should specify a direction of relationship between specific variables. I feel the "hypotheses" listed here should instead be presented as research questions which don't require as much specificity in their predictions.

Response: Thanks for kindly providing such insightful advice, which we agree and accept. Accordingly, we have replaced the hypotheses with a research question in the newly revised manuscript (Page 5, Line 21; Page 6, Line 1-3).

8. The aim should be presented before the hypotheses.

Response: According to your advice, we have replaced the hypotheses with the research question in the newly revised manuscript (Page 6, Line 3-6). Therefore, we keep the aim in the last sentence of the Introduction.

Materials and Methods-Participants and procedure

9. “The collection method was complied with the terms and conditions for the website.” Unclear what website is being referred to here. Do you mean the online questionnaire platform? Does that have an ethics compliance process? I'm just not sure how complying with terms and conditions for a website supports ethical practice in research. Which organisation or government body approves ethical procedures in research in China? Please clarify.

Response: We really apologize for this misleading and confusing statement, which we have deleted from the newly revised manuscript. What we wanted to express is that we complied with the terms and conditions, such as prohibition of the production, copying, publishing, and dissemination of pornographic, violent, obscene content and respect and protection of the personal privacy of all participants, which are required by the online questionnaire platform we used for the present survey study.

10. Delete the “received and” in “All completed questionnaires were received and checked for validity and completeness.”

Response: We have deleted “received and” from this sentence in the newly revised manuscript (Page 7, Line 1-2).

11. Materials and Methods-Measures

Change the word “proved” into “shown” in “The Chinese version of BSRS also included entertainment and physical practice, and was proved to have good reliability and validity in China.”

Response: We have change “proved” into “shown” in this sentence in the newly revised manuscript (Page 7, Line 8).

12.The lower the score was, the more regular the rhythm was. 

Response: We have deleted the “was” in the newly revised manuscript (Page 7, Line15-16).

13. Use the word "characteristics" instead of "impacts" and the words "rated on a scale of" instead of "graded to" for clarity in “Zung's self-rating depression scale (SDS) and Zung's self-rating anxiety scale (SAS) were used to evaluate the psychological impacts of each participant[17, 18]. There were 20 items in SAS and SDS, and each item was graded to 1-4 reflecting never, often, sometimes, and always.”

Response: We have changed the words “impacts” and “graded to” into “characteristics” and “rated on a scale of”, respectively, in the newly revised manuscript (Page 7, Line 19 and Page 7, Line 20).

14. Use the word "raw" instead of "rough" here for clarity in “The rough score was the sum of 20 questions multiplied by 1.25 and rounded to get the standard score.”and explain why scores are multiplied by 1.25.

Response: We have used the “raw” instead of “rough” in this sentence in the newly revised manuscript (Page 7, Line 21). It is a common practice to convert the results to a percentage scale in the psychometric instruments. Therefore, the raw score of SAS and SDS in the present study were multiplied by 1.25 as described in previous studies [19], which has been stated in the in the newly revised manuscript (Page 7, Lines 21; Page 8, Lines 1). 

References: Sun J, Sun R, Jiang Y, Chen X, Li Z, Ma Z, et al. The relationship between psychological health and social support: Evidence from physicians in China. PLoS One. 2020;15(1):e0228152. Epub 2020/01/30. doi: 10.1371/journal.pone.0228152. PubMed PMID: 31995601; PubMed Central PMCID: PMCPMC6988930.

15. Materials and Methods-Statistical analysis

Replace “scales of social rhythm scale”with "BSRS scales" in “Pearson correlation or Spearman correlation, where appropriate, was used to determine the correlation between the scales of social rhythm scale and the Zung’s scales.”

Response: We have replaced the “scales of social rhythm scale” with "BSRS scales" in the newly revised manuscript (Page 8, Line 16).

16. Add that statistical significance criteria were "corrected for multiple comparisons where appropriate" in “A P value of < 0.05 was considered statistically significant.”

Response: Thanks for your thoughtful suggestion. We have modified the sentence in the newly revised manuscript (Page 8, Lines 19-21) as “A P value of < 0.05 was considered statistically significant (corrected for multiple comparisons where appropriate).”

Results

17. Add the word “data” behind “invalid” and delete “the data analysis” in “Eighteen questionnaires with invalid were excluded the data analysis due to frivolous or incomplete answers, or the same option for different questions or orderly answers.”

Response: We have changed the sentence in the newly revised manuscript (Page 9, Lines 4) as “Eighteen questionnaires with invalid data were excluded due to frivolous or incomplete answers, or the same option for different questions or orderly answers.”

18. Adjusted for multiple comparisons? Please confirm. Some of the comparisons in Table 2 seem greater than the adjusted p value (e.g., Education, Chronic disease).

Response: Thanks for kindly providing such insightful advice. We have deleted “all” in the newly revised manuscript (Page 10, Line 7) as “Generally, there were significant differences in SGS-scale and EPE-scale among participants with different sociodemographic backgrounds including gender, age, educational background, annual income, health status, current occupation, and chronic disease status (P<0.05, Table 2).

19. An indication of effect size would help clarify the strength of these differences. Report Cohen's d statistic for all statistically significant comparisons in Tables 2, 3, and 4 to help with interpretation of the statistical outcomes.

Response: We have added a statement in the Statistical analysis section (Page 8, Lines 19-20), and reported the Cohen’s d statistic for all statistically significant comparisons in Tables 2, 3, and 4 of the newly revised manuscript.

20. Delete P<0.001 in “Additionally, SDS scores were significantly higher in medical students than in non-medical students (P<0.001).”

Response: We have deleted P<0.001 in this sentence in the newly revised manuscript (Page 14, Lines 9-10).

21. Add the “mean score” before “SGS-scale”, add the word “mean” before “SAS score” and “SDS score” in “Among the 5854 participants, SGS-scale and EPE-scale were 20.65±9.01 and 8.12±6.00, respectively, and SAS score and SDS score were 30.51±7.49 and 41.95±11.75, respectively.”

Response: We have modified this sentence in the newly revised manuscript (Page 16, Lines 6-7) as “Among the 5854 participants, mean scores on the SGS-scale and EPE-scale were 20.65±9.01 and 8.12±6.00, respectively, and mean SAS score and mean SDS score were 30.51±7.49 and 41.95±11.75, respectively. ”

Discussion

22. “lower or higher than average income” needs to be defined better. See my comment in the Abstract.

Response: We have changed it into “lower or higher than average income” in the newly revised manuscript (Page 18, Line 18-19; Page 19, Line 1-2).

23. Which hormones? I assume you mean female-specific hormones? Please clarify. “The endogenous circadian clocks can be reset by hormone signals, affecting the balance of the hypothalamus-pituitary-adrenal (HPA) axis, and making the HPA axis more responsive to stress than males.”

Response: Thank you for your kind suggestion. We have clarified this issue and modified the sentence in the newly revised manuscript (Page 19, Lines 7) as follow: “The endogenous circadian clocks can be reset by estrogen hormone signals in women, affecting the balance of the hypothalamus-pituitary-adrenal (HPA) axis, and making the HPA axis more responsive to stress than males.” 

24. “likely due to their weak competitiveness in the job market and inability to adapt to the new E-commerce models during the COVID-19 pandemic due to the isolation or even lockdown.” A reference is needed to support this statement.

Response: A reference has been added to support this statement in the newly revised manuscript (reference [26], Page 19, Line 19) 

Reference: Brenner MH, Bhugra D. Acceleration of Anxiety, Depression, and Suicide: Secondary Effects of Economic Disruption Related to COVID-19. Frontiers in psychiatry. 2020;11:592467. Epub 2021/01/02. doi: 10.3389/fpsyt.2020.592467. PubMed PMID: 33384627; PubMed Central PMCID: PMCPMC7771384.

25. Use the word "timely" instead of "timeless" to help with clarity. “Moreover, they did not have sufficient judgment while facing various, overloaded and timeless information on the COVID-19 pandemic, and easily felt stressed, depressive and anxious[27].”

Response: We have used the word "timely" instead of "timeless" in the newly revised manuscript (Page 19, Line 20).

26. Please re-write this highlighted section to improve clarity. It is unclear at the moment why people with high income are also the people with poor health status.- “Interestingly, in the present study, the people with high income were among those who mostly suffered from disrupted rhythms of entertainment during the COVID-19 pandemic. It has been shown that people with poor health status are likely to have basic diseases, weak immune function, and low ability to fight pathogens or adapt to a rhythm change[26], which may explain that people with poor health status suffered from disruption of social rhythm observed in the represent study.”

Response: Thank you for your thoughtful suggestion. We have rewritten this section to clarify the disrupted rhythm characteristics of different groups of people in the newly revised manuscript (Page 20, Lines 1-13) as follows: “Interestingly, both the people with high incomes and those with poor health status were mostly suffered from disrupted rhythms. On one hand, the disrupted rhythm in people with high incomes may be explained as follows: during the COVID-19 pandemic, the consumption expenditures of high-income groups dropped sharply by as much as 17%, while the expenditures of low-income families dropped by only 4%[28]. At the same time, entertainment venues that provide services for high-income groups, such as cultural activity centers, clubs and gymnasiums, golf playgrounds, luxury party venues, were mostly closed. All these changes would impact severely on the social rhythms of these groups of people. On the other hand, the disrupted rhythm in people with poor health status can be explained by the fact that their regular activities, such as walking, jogging, swimming and fitness, were significantly restricted since their poor health status with basic diseases, weak immune function, and low ability to fight pathogens or adapt to a rhythm change[29].

27. Correct the spelling in “Previously, Liu et al. [29]conducted a study on lifestyle and the health status of Chinee elderly, which included the elderly who lived in their own houses, with 20.6% of them being widows or widowers.”

Response: We have corrected the spelling of the word “Chinese” in the newly revised manuscript (Page 21, Line 5).

28. Has to be linked back to sleep - how does decreasing amplitude and phase advance of the circadian rhythm affect sleep. “Indeed, it has been reported that advanced age is characterized by a progressive decreasing amplitude and phase advance of circadian rhythmicity in overall biological functions,”

Response: Actually, this paragraph intends to address the reasons how COVID-19 pandemic impacted on the Chinese elderly through various internal and external factors. Therefore, we modified this section in the newly revised manuscript (Page 21, Lines 9-16) as follows “Indeed, it has been reported that advanced age is characterized by a progressive decreasing amplitude and phase advance of circadian rhythmicity in overall biological functions, including blood pressure, hormonal circadian secretions, and immune function changes[33]. The elderly could not go out for dancing after dinner, exercise in the morning, or visit old friends or relatives due to recommended social isolation. Furthermore, they were inherently at high risk for COVID-19 infection and easily generated negative emotions [34]. These various factors were prone to form a vicious circle.”

29. Deleted “that offers an explanation” in “The first factor is the social zeitgeber theory that offers an explanation. ”

Response: We have deleted it in this sentence in the newly revised manuscript (Page 22, Line 16 ).

30. Add "outcomes" after "mental health" in “Social stimuli under the stress of COVID-19 may affect the circadian behavioral programs by regulating the phase and period of circadian clocks (i.e. a zeitgeber action, either direct or by conditioning to photic zeitgebers) which leads to various adverse mental health.”

Response: We have added outcomes after “mental health” in this sentence in the newly revised manuscript (Page 23, Line 2).

31. I don't see this as a strong enough justification or explanation for the findings, especially as the reference source is based on studies in mice. Either drop this justification altogether or provide a stronger argument supported by more relevant research to humans. The third factor is SCN that serves as a master clock synchronizing the phase of clocks throughout the body. The disrupted rhythm in SCN leads to helplessness and behavioral despair, which increases depression and anxiety-related behaviors[36].

Response: We have deleted this point of view in the newly revised manuscript.

32. Add the word “respectively” after “social stress” to improve clarity in “Anxiety disorders and major depressive disorder have been associated with increased and blunted HPA axis reactivity to social stress, which engages the diverse biological pathways that bolster successful stress adaptation and promote stress resilience.”

Response: We have added the word “respectively” after “social stress” in this sentence in the newly revised manuscript (Page 23, Line 14).

33. Please correct the spelling in “Of course, other mechanisms may also connect social rhythm dysregulation with phycological impacts.” and add “, including circadian gene variations [39] and social jetlag [40].” at the end of this sentence. Then delete the section “A previous study reported that the 4/4 genotype of the circadian gene, PER3, was associated with anxiety symptoms in 804 Brazilian non-clinical young adults[39]. Mathew and his colleagues also found that social jetlag (a misalignment between sleep timing on the weekend and school week) was positively and independently associated with anxiety symptoms in adolescents[40].”

Response: We have modified this sentence as “Of course, other mechanisms may also connect social rhythm dysregulation with psychological impacts, including circadian gene variations [43] and social jetlag [44].” in the newly revised manuscript (Page 23, Line 19-20).

34. Please add a supporting reference in “2), spending 1 hour/day outdoors, walking for more than one hour a day and basking in the sun, which may regulate the body clock by the light-dark cycle;”.

Response: Thank you for your important advice. A supporting reference has been added in the newly revised manuscript (Reference 7, Murray G, Gottlieb J, Swartz HA. Maintaining Daily Routines to Stabilize Mood: Theory, Data, and Potential Intervention for Circadian Consequences of COVID-19. Can J Psychiatry. 2020:706743720957825. Epub 2020/09/11. doi: 10.1177/0706743720957825. PubMed PMID: 32909832.)

35. “Previous studies have shown that regular eating and consumption of various metabolites promote the periodic colonization of the intestinal bacteria on the mucosal surface, and alleviate various adverse mental problems through the intestinal flora-gut-brain axis[41];”- This isn't a circadian rhythm explanation - it deviates too far from the central themes of the paper and should be removed. 

Response: We have deleted this sentence in the newly revised manuscript.

---

## [Editor Report · Decision Letter 2]

14 Apr 2021

Disrupted rhythms of life, work and entertainment and their associations with psychological impacts under the stress of the COVID-19 pandemic: A Survey in 5854 Chinese people with different sociodemographic backgrounds

PONE-D-20-26146R2

Dear Dr. Chen,

We’re pleased to inform you that your manuscript has been judged scientifically suitable for publication and will be formally accepted for publication once it meets all outstanding technical requirements.

Kind regards,

Ben Bullock

Academic Editor

PLOS ONE
---

## [Editor Report · Acceptance letter]

7 May 2021

PONE-D-20-26146R2 

Disrupted rhythms of life, work and entertainment and their associations with psychological impacts under the stress of the COVID-19 pandemic: A Survey in 5854 Chinese people with different sociodemographic backgrounds 

Dear Dr. Chen:

I'm pleased to inform you that your manuscript has been deemed suitable for publication in PLOS ONE. Congratulations! Your manuscript is now with our production department. 

Kind regards, 

on behalf of

Dr. Ben Bullock 

Academic Editor

PLOS ONE